# Increased Early Cancer Diagnosis: Unveiling Immune-Cancer Biology to Explain Clinical “Overdiagnosis”

**DOI:** 10.3390/cancers15041139

**Published:** 2023-02-10

**Authors:** Bruce A. Wauchope, Brendon J. Coventry, David M. Roder

**Affiliations:** 1Molechecks Australia, 1284 South Road, Clovelly Park 5042, Australia; 2Discipline of Surgery, Cancer Immunotherapy Laboratory, University of Adelaide, Royal Adelaide Hospital, Adelaide 5005, Australia; 3Cancer Epidemiology and Population Health, Allied Health and Human Performance, University of South Australia, Adelaide 5000, Australia

**Keywords:** early, cancer, overdiagnosis, immune response, screening, public health

## Abstract

**Simple Summary:**

Earlier diagnosis is often advocated to prevent advanced cancer morbidity and death. Compelling evidence exists where this occurs. A major conundrum is the lack of supportive evidence for some cancers where more sensitive screening and early detection may not translate into reduced morbidity and mortality. Some early cancers do not advance to metastases and death, and thus may not endanger the patient during life, nor require treatment. Biological variants that are non-metastatic and non-lethal may exist. Distinguishing potentially ‘fatal’ cancers from ‘non-fatal’ cancers is fundamental for screening to selectively benefit, while avoiding unnecessary treatment. Immune control of cancer has evidential support, and may explain why some cancers progress, while others stay quiescent. The extent of immune system modulation of malignant behaviour could decisively influence cancer outcomes, including lethality. We advance an evidence-based, immune model, worthy of further research, potentially explaining cancer “overdiagnosis” and the susceptibility to recurrence, regression, and lethality.

**Abstract:**

Even though clinically small ‘early’ cancers represent many millions of cells biologically, when removed surgically, these often never recur or regrow, nor reduce the individual’s lifespan. However, some early cancers remain quiescent and indolent; while others grow and metastasize, threatening life. Distinguishing between these different clinical behaviours using clinical/pathological criteria is currently problematic. It is reported that many suspicious lesions and early cancers are being removed surgically that would not threaten the patient’s life. This has been termed ‘overdiagnosis’, especially in the sphere of cancer screening. Although a controversial and emotive topic, it poses clinical and public health policy challenges. The diagnostic differentiation between ‘non-lethal’ and ‘lethal’ tumor forms is generally impossible. One perspective gathering evidential support is that a dynamic balance exists between the immune response and malignant processes governing ‘lethality’, where many more cancers are produced than become clinically significant due to the immune system preventing their progression. Higher medical screening “diagnosis” rates may reflect lead-time effects, with more ‘non-progressing’ cancers detected when an early immune-cancer interaction is occurring. We present a model for this immune-cancer interaction and review ‘excess’ or ‘overdiagnosis’ claims that accompany increasingly sensitive diagnostic and screening technologies. We consider that immune tools should be incorporated into future research, with potential for immune system modulation for some early cancers.

## 1. Introduction

Increasingly sensitive screening and diagnostic technologies may produce higher cancer detection rates than indicated for many cancer types by morbidity/mortality rates from central registries. This may reflect lead-time effects, but if cumulative effects persist, it may reflect an ‘excess diagnosis’ (i.e., an ‘overdiagnosis’). This phenomenon is problematical for many clinicians and pathologists, in that many detected cancers may not have progressed to registered morbidity and mortality had they been left untreated. Overdiagnosis is defined as the detection of cancers from screening or diagnostic procedures that would not have otherwise gone on to cause illness or death if left untreated [1].

A cancer with continuous growth will, with few exceptions, ultimately be biopsied and reported in a population-based cancer registry. After accounting for lead-times and confounding variables, if the length-biased sampling appropriately reflects tumour growth rates [2], with time elapsing there should ultimately be a near equivalence of cumulative incidence between a screened representative sub-population and a population-wide registry, with the screened group having higher incidence rates initially due to increased early detection of cancers. After a lag, the earlier diagnosis rate should manifest some reduction in mortality rate.

If these cumulative incidence rates are not similar, with an elevation in the screened group, then questions may be raised about the potential for the overdiagnosis of ‘non-lethal’ cancers, which may raise ethical, cost, and other professional questions.

By comparison, effective cancer screening would entail the early detection of cancers *with* lethal potential or their precursors, leading to a reduction in morbidity and mortality [3]. Effective screening should translate into reduced cancer-specific mortality and age-adjusted incidence of advanced cancers.

However, overdiagnosis is reported in many clinical and screening environments, and may apply to skin screening for melanoma and squamous or basal cell carcinomas, mammography screening and examinations, bowel cancer screening, and PSA testing to detect prostate cancer.

In addition to the raw rates, relatively larger increases have been found between different types of lesions (e.g., more so for female breast in situ over invasive lesions). This applies to DCIS in relation to breast mammography, and to other in situ cancers—prostate, colon, and squamous cell cancers (SCC) of the gastrointestinal tract, genital tract and skin, basal cell cancer (BCC) types, and cutaneous melanomas [2,4]. In Korea, thyroid cancer has a very high overdiagnosis rate [5]. In the UK, uterine, prostate, oral and thyroid cancer incidence and mortality data are suggestive of overdiagnosis and trends in melanoma and kidney suggest overdiagnosis with some underlying increase in true risk, while cervical and breast cancer trends may also reflect improvements in treatment and earlier diagnosis [6].

Melanoma-screened sub-populations have shown both higher detection rates and higher in situ to invasive rates than expected from the population-based registry data, without a lower-than-expected mortality arising for melanoma [1,4,7].

There have been long-standing questions raised about the rise in melanoma incidence and whether non-lethal, ‘non-metastasizing’ forms of melanoma are being diagnosed. Thus, a vocabulary that includes overdiagnosis, asymptomatic reservoirs of ‘indolent’ conditions, ‘dormant’ and ‘non-metastasizing’ forms of melanomas have emerged [7,8,9,10,11,12].

It has been proposed that overdiagnosis might be the consequence of common screening tests [3]. It also is even more likely to occur and to increase in diagnostic settings with the advent of increasingly sensitive diagnostic technologies. There have been suggestions that lower thresholds to perform biopsy have occurred with clinicians, and with pathologists, shifting diagnostic thresholds, with these together leading to an increase in case detection rates, giving an apparent impression of success. Financial incentives have been suggested also. Together they all may drive a screening cycle incentive [7].

The clinical significance of cancers detected during screening becomes more questionable if there is not a corresponding reduction in morbidity and mortality. In the context of increased notifications, contrary public health recommendations have then been made to either increase screening; restrict it to ‘high risk’ subgroups; or in some cases, to discontinue screening altogether [7].

In Australia, it has been estimated that 18% of all cancers diagnosed in women and 24% of all cancers diagnosed in men are overdiagnosis events [13]. Elevations on this or similar scale would impose a major cost burden and present a significant clinical and public health challenge.

## 2. Overdiagnosis and Pathology Classification

It has been proposed that the overdiagnosis rates may be due to changes in pathology diagnostic thresholds, combined with increased screening.

Pathologists, when presented with slides from 20 years ago, increased the diagnosis rates of melanomas: 14% of severely dysplastic lesions were converted to melanoma [12]. While this is a significant change it does not account for the very large increase in melanoma diagnosis.

Similar issues have been raised with the 15-times increase in thyroid cancer, which has driven a change in thyroid pathology classification. These recent changes in classification have reduced the high rate of thyroid cancer by 1/6 [14]. Again, these changes made to address the 15-times increase in thyroid cancer rates do not solve or answer this problem, merely alter the definition.

## 3. Overdiagnosis and Cell Growth Rates

Underlying this phenomenon of overdiagnosis is an understanding that the biological behaviour of cancers found on screening is not consistent. Some cancers found on screening, while being a legitimate diagnosis with pathological verification, may not progress to be clinically significant. The biological behaviour and cell growth rates are not necessarily shared by cancers. This is supported by autopsy studies finding rates of ‘hidden’ or ‘occult’ cancers that have not been causing morbidity, mortality, nor been diagnosed during life.

Overdiagnosis has been attributed to detecting a reservoir of slow growing cancers that do not, or even possibly cannot, metastasise and are ‘benign’ in clinical terms. The published prerequisites of excess or overdiagnosis are a reservoir of silent disease and a screening or detection activity that leads to detection of subclinical disease within the reservoir [3].

This is shown in the diagram associated with overdiagnosis (Figure 1).

This schematic illustrates the large variability in growth rates and lethal potential of malignant cells. Overdiagnosis occurs when screen-detected cancers are either non-growing or so slow-growing that they would never cause medical problems before death from other causes.

In this model of overdiagnosis, the inherent growth rate of the cancer is important. This growth rate it is based on isolated cell growth alone, without an interaction or reactive expression from the body within which the cancer is growing.

In assessing this growth rate of tumours and their metastatic capacity, markers may be beneficial, such as gene expression signatures that relate to proliferation and metastasis of more aggressive tumours and gene expression signatures related to ageing and senescence of more indolent cancers [15,16,17].

## 4. Reflecting on Excess- or Overdiagnosis

With the potential overdiagnosis conundrum, questions emerge. Is overdiagnosis due to the properties of the cancer cell types and cell behaviours alone, or is their rate of growth disconnected from interaction with the rest of the body? Are cancer cell types and growth rates alone enough to explain it all? We raise other questions.

Does the cancer exist alone, or is it in relationship with the rest of the body and the immune system? How is the immune system involved in the cancer microenvironment and cancer growth? Can the immune system affect the growth of cancers? In other words, can the immune system restrict the growth of the cancer, or/and can it increase the growth of the cancer? In addition, is the immune system plastic? That is to say by how much can it alter its profile, or is it fixed and static? If the immune system alters, can it affect the growth of cancer? Can altering the immune system cause a change in cancer behaviour and outcome? These questions are helped by background data. It is well established that immunosuppression of ‘healthy’ people leads to an at least 3-times increase in cancer (excluding sun exposed sites), and 100-times increase in NMSC [18,19,20,21]. On the treatment side, the use of immune checkpoint inhibitors has transformed cancer survival, but only for a proportion of cancer types (some 1–50%) [22].

In addition, many clinicians are familiar with rare but striking instances of spontaneous tumour regression, a process where some cancers spontaneously disappear, potentially as a result of immunological processes. Although explaining it remains an enigma, it may be more common than is appreciated, perhaps more so with ‘early’ cancers. At a cellular level, immuno-editing is a well-established process in cancer research and a basis for research into cancer therapies.

In summary, evidence shows that the immune system can affect formation, progression, and mortality from cancer. In this, we are looking at more than isolated cancer cell growth rates: an interaction between the cancer and the immune system.

Holding together both the controversial understandings of putative overdiagnosis and the immune system’s effect on cancer development, progression, and survival, we propose using these together as a window into clinical cancer pathophysiology. Thus, we propose that overdiagnosis can also occur by detection in the early phases of cancer development, in the context of immune-tumour interactions.

## 5. Cancer and the Immune Dynamic

The role of the immune system in managing infections is well-accepted, but in cancer the immune role has been less clear for decades. The immune system is critical for functional integration and cellular co-operation. It is involved with tissue development, homeostasis, defence, repair, and clearance of debris; returning to tissue development, maintenance and homeostasis in an ongoing circle. This function is well beyond recognising and removing ‘non-self’ cells.

This circular process of tissue development, homeostasis, defence, repair, and clearance of debris has both *anabolic* and *catabolic* stages. It requires the development and recognition of ‘self’ and the recognition and defence from ‘non-self’, which may not be fixed—for example, in immunotherapy, autoimmunity is associated with tumour regression and rejection.

Clarifying this broader role of the immune system helps to understand these enigmas in cancer research. Both the anabolic and catabolic features of the immune system are important when assessing the research on cancer because research shows that the immune system can promote or restrict the growth of cancers. While this seems paradoxical, it is coherent if considered within a broader understanding of the immune system functions.

Pradeu [23] has outlined three ways that the immune system manages these roles:Filtering entry. This is at points of entry to the body.Filtering presence. This process of filtering occurs by monitoring the rate of change. This has been historically called ‘immunosurveillance’.Promotion of cooperation between the body elements.

Recently, cancer treatments, immune therapies, inhibitors of CTLA-4, PD-1, and PD-L1, have caused a paradigm shift in our understanding of the integral immune system role in cancer, and have re-focused medical thinking on a door that goes back through recent history, to at least William Coley, as perhaps the ‘Father of Immunotherapy’ [24]. This shift in thought continues, with increasing clinical recognition that the immune system, in addition to managing infections, is front and centre to normal tissue growth, repair, and wound healing. With both growth and healing having immunological inputs and controls, it is evident that similar immunological phenomena regulate tissue growth and repair in the cancer micro-environment. The capacity for the immune system to promote tumour growth [25] is gaining increased clinical recognition.

In the tumour microenvironment, the stroma, particularly its immune components, interact with the tumour affecting its growth and progression [26]. Displaying the understanding that the immune system can promote or restrict a cancer, various immune cell profiles in the cancer microenvironment can be associated with promotion or restriction of tumour development. Tumour associated macrophages (TAMs), if polarised to M2 are associated with tumour growth and metastasis [25]. The entire immune cellular ‘household’ can affect cancer growth in different ways [26,27,28,29,30,31,32,33,34,35,36].

Indeed, the balance of the immune system can determine the clinical outcome in many diseases, including in cancer. “Data from large clinical studies demonstrate indeed that a robust infiltration of neoplastic lesions by specific immune cell populations, including (but not limited to) CD8+ cytotoxic T lymphocytes, Th1 and Th17 CD4+ T cells, natural killer cells, dendritic cells, and M1 macrophages constitutes an independent prognostic indicator in several types of cancer. Conversely, high levels of intratumoral CD4+CD25+FOXP3+ regulatory T cells, Th2 CD4+ T cells, myeloid-derived suppressor cells, M2 macrophages and neutrophils have frequently been associated with dismal prognosis” [37]. Immune system control may be postulated as the influential arbiter of metastasis, disease progression, and survival. Immunotherapy studies and immune system understanding suggest an influential role of immune mechanisms in cancer control, previously less appreciated [38]. There appears to be no definitive data at present to provide insight into any potential difference between immune cellular infiltration at the immunosurveillance stage in very early cancer development and infiltration of the immune system in more developed indolent or pathogenic lesions to help distinguish them.

In the light of the immune interactions affecting growth in the microenvironment and also metastasis, we are suggesting the immune-tumour interaction critically affects growth outcomes, affecting rates of tumour growth, affecting its capacity to be indolent or pathogenic, and in some cases its disappearance and regression.

Thus, in addition to exploring the biology of tumour growth rates and their genomic, transcriptomic, and proteomic expression, we suggest the interactions between the immune system and the cancer, are foundational, dynamic, and can be cognised in relationship terms.



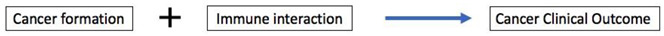



In the immune system—cancer relationship, both the global immune system and the local tumour microenvironment are critical. Evidence has emerged from cancer immunology and immunotherapy leading to an emerging proposal that the immune system may be important, perhaps central, to the issue of overdiagnosis. The immune-cancer dyad is critical to clinical outcome.

With this background, we return to the three observed outcomes in screening:Increased diagnosis ratesIncreased in situ/invasive ratiosIncreased claims of overdiagnosis

Our proposition is that these three outcomes may also logically be biological manifestations of the immune—cancer relationship in the earlier stages (or maybe earliest) of cancer development.

## 6. A Model for Overdiagnosis and the Immune System through Melanoma

Incorporating these ideas, we present here a composite model based on the example of melanoma screening, which examines the interaction between early tumour formation and the immune response, immuno-editing, which may shed some light on the issues mentioned above and can explain what we see clinically (Figure 2).

Melanoma tumorigenesis involves proliferating aberrant melanocytes with reduced contact inhibition, and with growing dis-cohesiveness in a multicellular organism. The immune system surveillance detects the tumour at (A). If the immune system is ‘tumour-promoting’, the upper grey part of the diagram (B) becomes operational. The tumour is then facilitated in its growth, as occurs in proliferative wound healing. In the tumour microenvironment, innate immune cells such as macrophages can be associated with tumour progression [25]. Clinically, as an example, this is represented by rapidly growing nodular melanomas. These penetrate deeply with less sideways spread, and while accounting for a small percentage of melanoma overall, they contribute 30% of deaths. Early detection < 3 mm diameter is preferred.

If the immune system is ‘tumour-restraining’, the lower, orange part of the diagram (C) becomes operational. Detailed screening at (D) detects melanomas in the earlier stages of the immune—tumour interaction.

The melanoma enters a regression phase, with as per the diagram labelling (1) elimination, (2) equilibrium, or (3) escape, as described:(1)**Elimination of the tumour**: Immune ‘regression’ associated with lymphocytes may remove all histopathological evidence of melanoma. Clinically this is seen as spontaneous regression, or primary regressing/regressed melanoma.(2)**Equilibrium**: The immune system has not removed the tumour, but has restrained it. The tumour and the immune system may reach a steady state. Clinically, this is found in the post-mortem data of people dying with, but not from, cancers. This may be much of the reservoir in overdiagnosis.The cancer is evolving, and the immune profile is dynamic, thus this situation can shift. The immune system can overcome the tumour causing regression and elimination. Conversely, the tumour may overcome the immune restraint, and ‘escape’.(3)**Immune escape**: The cancer may initially be restrained by the immune system, but then overcome it. If the immune system eliminates the primary tumour after metastases are released, metastatic secondaries with no known primary occurs. Clinically, this is known as ‘occult melanoma or ‘melanoma of unknown primary origin’ which occurs in about 3% (range 1.2–31%) of advanced melanomas [39]. This also occurs in some other cancer types.

The 3 es of immunoediting of the cancer by the immune system, elimination, equilibrium, and escape, can overlap.

Detailed screening at (D) detects melanomas in the earlier stages of the immune—tumour interaction. Use of dermatoscopy with SDDI (sequential digital dermatoscopic imaging), and TBP (total body photography) enables early detection and will give a lower benign/malignant ratio on excision [40]. Screening at (D) will give

A higher Melanoma detection rate and an even larger increase in the detection rate of in situ melanomas than would otherwise be present in a population-based cancer registry. This gives,An increased MIS/Invasive ratio. The raised melanoma and in situ detection despite no reduction in mortality, to give;Relative overdiagnosis.

## 7. Discussion

In summary, the controversy of over-screening, increased detection rates and calls of overdiagnosis, may provide us with a unique window that directs clinical thinking to the ‘immune—cancer dyad’, and may provide an instructive opportunity to better understand and even treat malignancy more effectively, including more cost-effectively. Indeed, if immune system control of cancer exists in all or most cases of cancer, then overdiagnosis may actually be a reflection of the extent of better immune control of the capacity of cancer cells to behave non-metastatically (or otherwise).

Important unresolved questions of immense clinical and public health significance arising from this model and an ‘immune—cancer dyad’ are:Which cancers will (or will not) progress and become clinically significant?Which cancers will translate through to metastasis and mortality?Which immune profiles are operating on a tumour and at what time?How might we alter immune profiles in a clinically useful way?

## 8. Conclusions

Cancer overdiagnosis with screening may also have an immune basis; this is supported by accumulating evidence.Cellular/immune profiling is currently lacking to identify which lesions will be held in check by immunological defence, or will be eliminated, or progress to metastasis.There are no current available clinical or pathology-based means of quantifying the immune—tumour interaction when deciding on need for treatment.The immune—tumour interaction should be an important focus of increased research to better understand and improve cancer control. Skin, being an external organ, is ideally accessible for this research pursuit.

It is considered that we are at some risk of becoming better and better at detecting early cancers that do not threaten or endanger the patient during their lifetime—with current clinical indecision as to which cancers will become invasive or metastatic, and which will not. The arbiter may indeed not just be the cancer cells themselves which has occupied so much attention to date, but rather in the dynamic behaviour and strength of the host immune system response. We present a model for this immune-cancer interaction and review the ‘excess’ or ‘overdiagnosis’ dilemma arising from increasingly sensitive diagnostic cancer screening technologies, suggesting immune tools should be incorporated into future research, and even potential immune system modulation of early cancers. Further research is required to define the distinction between cancers which can progress to fatality, and those that cannot or do not. In that way, a more accurate diagnosis may well be obtainable to reduce any excess in diagnosis of cancer that is not associated with clinical significance, including mortality. Our model may assist this.

## Figures and Tables

**Figure 1 cancers-15-01139-f001:**
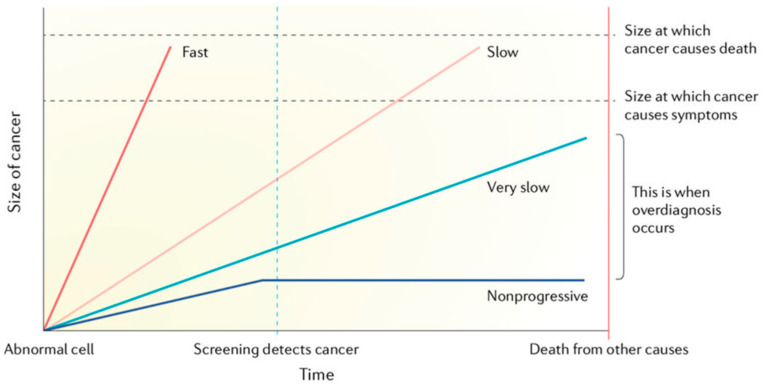
Slow versus rapid progressors—unpredictable tumour growth trajectory. (From Welch [1], as adapted by NCI Division of Cancer Prevention [3]).

**Figure 2 cancers-15-01139-f002:**
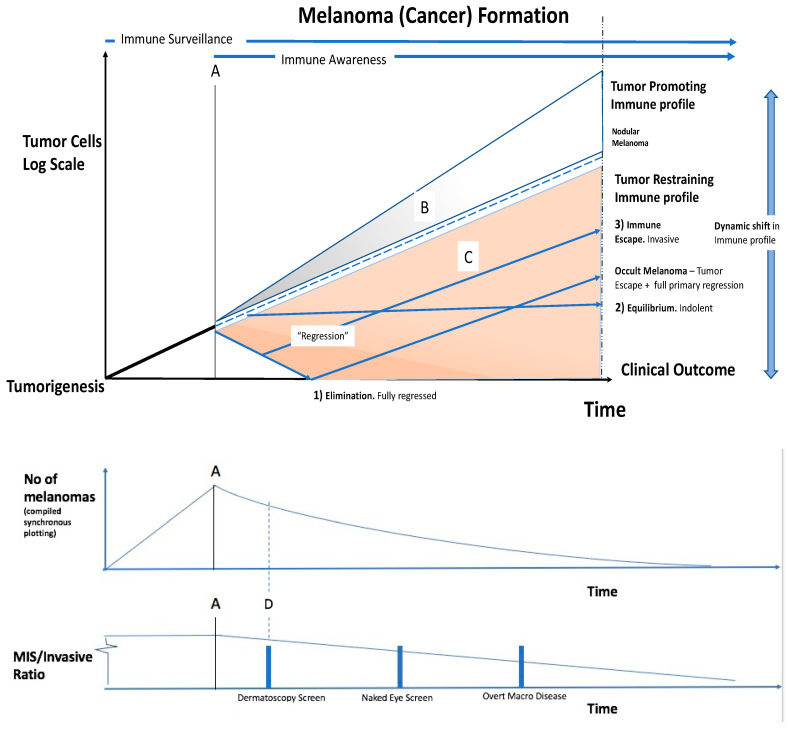
Model of the immune—cancer interaction.

## Data Availability

All data can be found in the text.

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
