# Peer review of "Increased Early Cancer Diagnosis: Unveiling Immune-Cancer Biology to Explain Clinical “Overdiagnosis”"

_cancers, 2023, doi:10.3390/cancers15041139_

Round 1

Reviewer 1 Report

The perspective written by Wauchope and his colleagues is a well written article and easy to read. This is an area of research I am deeply interested in. Overdiagnosis and Overtreatment as they are known to the public heath officials are of health concern. Millions of dollars are spent each year on treating patients with indolent or benign disease that do not require any intervention and do not affect the outcome. Authors have meticulously listed the cancer types that are usually over-diagnosed, e.g., prostate, breast, renal, thyroid, etc.  Authors have explained the epidemiological in support of overdiagnosis, i.e., increase in screen-detected cases without any change in mortality. There have been a number of articles related to this phenomenon and fortunately, authors have cited those articles. Having said that, the perspective can benefit from some real data, especially in the immune dynamic. Most of the points raised in this article is by conjecture only and lack supporting data. Here are some suggestions for the improvements:

·       Provide data, if any, on as to how early immune infiltration takes place in indolent or benign lesions

·        Has anyone identified “immune privileged’ versus “immune susceptible” subpopulation of cells based on single cell characterization and how do they affect the progression of indolent lesions? What kind of imbalance needs to exist between these two types of cells?

·       Provide data on immune components of stroma, if available, to support the growth and progression

·       I agree that the entire immune cellular ‘household’ can affect cancer growth in different ways, but the question is in  its relevance to indolent lesions. Author should provide a statement or two on this very relevant question providing scientific underpinning behind the progressor versus non-rogressor indolent lesions.

In summary, the perspective is well written and should be considered for publication.

Author Response

Reviewer 1

  1. Provide data, if any, on as to how early immune infiltration takes place in indolent or benign lesions.

Author response:  Prior to immune infiltration we have immune surveillance.  If, as is understood, the immune system is involved in tissue development and maintenance of tissue homeostasis in addition to defence repair and clearance, then surveillance is intimately involved in normal tissue growth.  Data on the transition from ‘normal’ tissue immune infiltration to the very earliest stages of ‘malignant’ development are intensely difficult to obtain and define and we are not aware of any human data on this point. I have not data on the shift from surveillance to infiltration, and how this then has either a controlling effect or promoting effect on growth of nevi.

  1. Has anyone identified “immune privileged’ versus “immune susceptible” subpopulation of cells based on single cell characterization and how do they affect the progression of indolent lesions? What kind of imbalance needs to exist between these two types of cells?

Author response:  We like this question - it is a very good one, regarding the cellular input into the “cancer immune dyad” on both sides of the equation in indolent lesions and how it might determine the effector response and consequent regression, stability or progression. This immunological information is not available to our knowledge at present for single cell analysis, let alone the more relevant whole intact tissue situation. We are considering malignancy at its very earliest stages initially, and then as invasiveness occurs.

To emphasise, we are considering the immune cell:cancer cell interaction dyad as a constantly dynamic relationship which adds yet another layer of complexity to static tissue (static ‘snap-shot picture’ at one time point) analysis or isolated single cell analysis.  In this paradigm the immune system is capable of driving or inhibiting the progression of the cell growth - but in a continuously changing dynamic manner (more like a ‘movie’ over time). A view at one point in time might therefore not reflect this fully.   

This is seen in the tumour microenvironment where M1 macrophages promote cytotoxic T-cell (CTL) (CD8) effectiveness and M2 macrophages inhibit CTL effectiveness.

Continuing with your question we can ask: “in indolent lesions, have we identified cell subpopulations that affect this immune: cancer dyad such that the cell interaction during immunosurveillance promotes (for example ) either a M1 or M2 response. 

Answer:  No, not at this stage in indolent lesions. 

In more advanced invasive cancers (as they go from in-situ to invasive) this is well established i.e. pancreatic ductal adenocarcinomas, the ras oncogene may drive an inflammatory program that establishes immune privilege (Vonderheide, R.H. and L.J. Bayne, Inflammatory networks and immune surveillance of pancreatic carcinoma. Curr Opin Immunol, 2013. 25(2): p. 200-205)

In addition the removal of cells from the stroma and its dyadic communication, has proved that definitional differentiation between ‘indolent’ and ‘cancerous/ progressive’ can be difficult – ie in breast cancer.

“Despite vast tumor cell-intrinsic differences, aggressive and indolent tumor cells are functionally indistinguishable once isolated from their local niche, suggesting a role for non-tumor collaborators in determining aggressiveness” (Sinha, V.C., et al., Single-cell evaluation reveals shifts in the tumor-immune niches that shape and maintain aggressive lesions in the breast. Nat Commun, 2021. 12(1): p. 5024).

  1. Provide data on immune components of stroma, if available, to support the growth and progression

Author response:  Data from large clinical studies demonstrate robust infiltration of neoplastic lesions by specific immune cell populations, including (but not limited to) CD8+ cytotoxic T lymphocytes, Th1 and Th17 CD4+ T cells, natural killer cells, dendritic cells, and M1 macrophages, constituting an independent prognostic indicator in several types of cancer. Conversely, high levels of intratumoral CD4+CD25+FOXP3+ regulatory T cells, Th2 CD4+ T cells, myeloid-derived suppressor cells, M2 macrophages and neutrophils have frequently been associated with a dismal prognosis.  (Senovilla, L., et al., Trial watch: Prognostic and predictive value of the immune infiltrate in cancer. Oncoimmunology, 2012. 1(8): p. 1323-1343)

It is also noted that this appears to vary with the tumour histological type, so the variability renders pattern identification problematic to some degree, even in invasive cancers, let alone very early cancers at the establishment stages. None of this, however, diminishes the importance of the reviewer’s comment/ question above.

  1. I agree that the entire immune cellular ‘household’ can affect cancer growth in different ways, but the question is in its relevance to indolent lesions. Author should provide a statement or two on this very relevant question providing scientific underpinning behind the progressor versus non-progressor indolent lesions.

Author response:  The penetrating first question on immune infiltration in indolent or benign lesions and the role in progression interrogates the vital ‘switches’ that must occur to determine whether ‘indolence’ or ‘progression’ prevails. We do not have data on this and we are unaware of any that exists, but would be appreciative if there is some? Indeed, in the dynamic model we propose, the whole process is not static at any point, but is capable of change at several levels determined by the immune:cancer dyad and the balance that prevails over time - so obtaining in-vivo dynamic data may prove scientifically difficult using current methods.

Reviewer 2 Report

In this perspective, authors presented a based-evidence immune model, potentially explaining cancer “overdiagnosis” and the susceptibility to recurrence, regression, and lethality.

The Title of the manuscript is concise and relevant. Introduction is quite comprehensive and highlighted work importance. Overall, the perspective is nicely written.

Before proceeding further, I expect the authors to thoroughly proofread the document and fix all grammatical and typographical errors (like L221 etc).

Minor suggestions:

1)      L76, L192, L197: kindly mention reference numbers together in single square bracket, separated with commas and with ‘-‘ (ex:22-24)

2)      L82/84: if reference points from the same reference lined up back to back, then avoid mentioning the same reference again and again. One time denotation is enough.

3)      L86, L79: brackets should come before fullstop. Maintain writing consistency throughout the paper.

4)      References: maintain theme font consistent throughout the paper. References font is different from rest of the paper.

5)      L246, L249, L257: replace numbers with small case alphabets, to avoid confusion.

Author Response

Reviewer 2

Minor suggestions:

1)      L76, L192, L197: kindly mention reference numbers together in single square bracket, separated with commas and with ‘-‘ (ex:22-24)

2)      L82/84: if reference points from the same reference lined up back to back, then avoid mentioning the same reference again and again. One time denotation is enough.

3)      L86, L79: brackets should come before fullstop. Maintain writing consistency throughout the paper.

4)      References: maintain theme font consistent throughout the paper. References font is different from rest of the paper.

5)      L246, L249, L257: replace numbers with small case alphabets, to avoid confusion.

Author response:  Thank you.  These have been attended to in the revised version.

Reviewer 3 Report

This is an interesting and useful exposition on the hypothesis that some overdiagnosis can be explained by considering the immune response to cancer - slowing, halting, or reversing growth. While I agree that cell growth rates and immune response are likely to be an important component of overdiagnosis, this article ignores other contributors and explanations such as the mis-classification problems. I suggest this broader context of multiple explanations needs to be set out in the Introduction and Discussion.

Major comments

1/ I noticed thyroid cancer is not mentioned anywhere in the article, but has probably the highest % of overdiagnosis, with a 15-fold increase seen in S Korea from screening (Ahn 2014). It would be helpful to have a more complete list of the overdiagnosed cancers, eg Oke 2018 and/or Glasziou 2020.

2/ Page 3. While I agree that cell growth rates are an important component of overdiagnosis, this ignores the classification problem. Pathologists disagree about whether the histology represents “cancer” and what grade it is. The classification also changes, eg thyroid cancer classification changed a few years ago, which reduced the % designated as "cancer" by about 1/6 (Nikiforov, 2016) - substantial but not enough to explain the S Korea increase.

3/ Figure 1. The attribution for this Figure seems to be missing (original is Gil Welch from Dartmouth, but I think this version is from NCI Division of Cancer Prevention's adaptation)

4/ Page 5: The statement: "leads to the proposal that the immune system is central to the issue of overdiagnosis." is not justified. The immune system plays some role, but the authors have not quantified the relative roles of the other mechanisms for overdiagnosis, so cannot claim immune system is central.

5/ Page 7. The quotes around “overdiagnosis” and “early” throughout the document become irritating. It is clear the authors accept the concept and quantification of overdiagnosis, and a providing an explanation rather than denying the existence of overdiagnosis.

6/ Minor: I noticed that David Roder is misspelled as “David Order” in the title/authorships, so I wonder if he had read the final version?

References

  1. Ahn HS, Kim HJ, Welch HG. Korea's thyroid-cancer "epidemic"--screening and overdiagnosis. N Engl J Med. 2014 Nov 6;371(19):1765-7. doi: 10.1056/NEJMp1409841. PMID: 25372084.
  2. Oke JL, O'Sullivan JW, Perera R, Nicholson BD. The mapping of cancer incidence and mortality trends in the UK from 1980-2013 reveals a potential for overdiagnosis. Sci Rep. 2018 Oct 2;8(1):14663. doi: 10.1038/s41598-018-32844-x. PMID: 30279510; PMCID: PMC6168593.
  3. Glasziou PP, Jones MA, Pathirana T, Barratt AL, Bell KJ. Estimating the magnitude of cancer overdiagnosis in Australia. Med J Aust. 2020 Mar;212(4):163-168. doi: 10.5694/mja2.50455. Epub 2019 Dec 19. Erratum in: Med J Aust. 2020 Apr;212(6):253. PMID: 31858624; PMCID: PMC7065073.
  4. Nikiforov YE, Seethala RR, Tallini G, et al. Nomenclature Revision for Encapsulated Follicular Variant of Papillary Thyroid Carcinoma: A Paradigm Shift to Reduce Overtreatment of Indolent Tumors. JAMA Oncol. 2016 Aug 1;2(8):1023-9. doi: 10.1001/jamaoncol.2016.0386. PMID: 27078145; PMCID: PMC5539411.

Author Response

Reviewer 3

Comments and Suggestions for Authors

This is an interesting and useful exposition on the hypothesis that some overdiagnosis can be explained by considering the immune response to cancer - slowing, halting, or reversing growth. While I agree that cell growth rates and immune response are likely to be an important component of overdiagnosis, this article ignores other contributors and explanations such as the mis-classification problems. I suggest this broader context of multiple explanations needs to be set out in the Introduction and Discussion.

Major comments

1/  I noticed thyroid cancer is not mentioned anywhere in the article, but has probably the highest % of overdiagnosis, with a 15-fold increase seen in S Korea from screening (Ahn 2014). It would be helpful to have a more complete list of the overdiagnosed cancers, eg Oke 2018 and/or Glasziou 2020.

Author response:  Agreed.  We took the Western approach to overdiagnosis from Srivastava 2019 (Srivastava, S., et al., Cancer overdiagnosis: a biological challenge and clinical dilemma. Nat Rev Cancer, 2019. 19(6): p. 349-358.) 

A more global approach is an improvement, and we have included this in.

2/  Page 3. While I agree that cell growth rates are an important component of overdiagnosis, this ignores the classification problem. Pathologists disagree about whether the histology represents “cancer” and what grade it is. The classification also changes, eg thyroid cancer classification changed a few years ago, which reduced the % designated as "cancer" by about 1/6 (Nikiforov, 2016) - substantial but not enough to explain the S Korea increase.

Author response:  Thank you we have included this aspect better. The contributions of classification component, sampling rates, UV exposure has been raised by Welch, 2021 (Welch, H.G., B.L. Mazer, and A.S. Adamson, The Rapid Rise in Cutaneous Melanoma Diagnoses. N Engl J Med, 2021. 384(1): p. 72-79.). 

Welch reported the data by Frangos (Frangos, J.E., et al., Increased diagnosis of thin superficial spreading melanomas: A 20-year study. J Am Acad Dermatol, 2012. 67(3): p. 387-94.) where 4 out of 29 severely dysplastic nevi were upgraded to melanoma by pathologists reviewing slides from 20 years ago. 

This shift in approx. 14% of severely dysplastic lesions to melanoma when combined with a doubling of sampling, does raise the rates but not to the 6 times required.  Welch suggests that financial self-interest may be involved here.  While this driver is significant for the USA, there is not the same self-interest involved in combined biopsy and self-reporting in the UK or Australia for example, as the biopsy and pathology are separated. 

Like the partial effect of the classification changes in thyroid cancer in S Korea there is significant room to think of other inputs.  We are suggesting that screening into the earlier stage of the cancer / immune dyad is finding many lesions that would not have progressed to become clinically significant.

3/  Figure 1. The attribution for this Figure seems to be missing (original is Gil Welch from Dartmouth, but I think this version is from NCI Division of Cancer Prevention's adaptation)

Author response:  Many thanks. The figure is attributed now.  Yes, this figure is taken from Srivastava 2019 (Srivastava, S., et al., Cancer overdiagnosis: a biological challenge and clinical dilemma. Nat Rev Cancer, 2019. 19(6): p. 349-358.), which referenced from Welch.

4/ Page 5: The statement: "leads to the proposal that the immune system is central to the issue of overdiagnosis." is not justified. The immune system plays some role, but the authors have not quantified the relative roles of the other mechanisms for overdiagnosis, so cannot claim immune system is central.

Author response:  Yes true, on reflection this was perhaps overstated and has been changed to a strongly stated conditional statement.

5/ Page 7. The quotes around “overdiagnosis” and “early” throughout the document become irritating. It is clear the authors accept the concept and quantification of overdiagnosis, and a providing an explanation rather than denying the existence of overdiagnosis.

Author response:  Thankyou agreed - this has been rectified.

6/ Minor: I noticed that David Roder is misspelled as “David Order” in the title/authorships, so I wonder if he had read the final version?

Author response:  This was not yet detected, but almost certainly was the result of ‘spellchecker’ at the final version just before submission.  Thankyou. David Roder did in fact read/ edit and his name was correct in penultimate version.

Round 2

Reviewer 3 Report

THe authors have responded well to my previous comments, and the paper is now OK to publish